# Spherical Channels for Modeling Atomic Interactions

**C. Lawrence Zitnick**[1], **Abhishek Das**[1], **Adeesh Kolluru**[2], **Janice Lan**[1], **Muhammed Shuaibi**[2],
**Anuroop Sriram**[1], **Zachary Ulissi**[2], **Brandon Wood**[1]
[1] Fundamental AI Research at Meta AI
[2] Carnegie Mellon University

## Abstract

Modeling the energy and forces of atomic systems is a fundamental problem in computational chemistry with the potential to help address many of the world's most pressing problems, including those related to energy scarcity and climate change. These calculations are traditionally performed using Density Functional Theory, which is computationally very expensive. Machine learning has the potential to dramatically improve the efficiency of these calculations from days or hours to seconds.

We propose the Spherical Channel Network (SCN) to model atomic energies and forces. The SCN is a graph neural network where nodes represent atoms and edges their neighboring atoms. The atom embeddings are a set of spherical functions, called spherical channels, represented using spherical harmonics. We demonstrate, that by rotating the embeddings based on the 3D edge orientation, more information may be utilized while maintaining the rotational equivariance of the messages. While equivariance is a desirable property, we find that by relaxing this constraint in both message passing and aggregation, improved accuracy may be achieved. We demonstrate state-of-the-art results on the large-scale Open Catalyst 2020 dataset in both energy and force prediction for numerous tasks and metrics.

## 1 Introduction

Modeling the properties of atomic systems is a foundational challenge in computational chemistry and critical to advancing technologies across numerous application domains. Notable applications include drug discovery [32, 39] and the design of new catalysts for renewable energy storage to help in addressing climate change [48, 33]. For catalyst discovery, new materials are currently evaluated using Density Functional Theory (DFT) that can estimate atomic energies and forces but is computationally very expensive; taking hours or days to evaluate a single material. Machine Learning (ML) has the potential to approximate DFT and dramatically speed up these calculations, allowing for high throughput screening of new materials to help address some of the world's most pressing challenges.

Our goal is to approximate DFT calculations using ML. An ML model takes as input a set of atom positions and their atomic numbers. As outputs, a model calculates the structure's energy (or other properties) and the per-atom forces, i.e., the forces exerted on each atom by the other atoms. A common approach to this problem is to use Graph Neural Networks (GNNs) [19, 47] where each node represents an atom and the set of nearby atoms as edges [37, 17, 24, 35, 38, 45, 31, 13].

A key challenge in network design is balancing the use of model constraints. When modeling atomic systems, a common constraint is SO(3) rotation equivariance [43, 3, 1, 42, 34, 36], *i.e.*, if the atomic system is rotated, the energies should remain constant and the atomic forces should similarly rotate. While this provides strong priors on the model to help in generalization, especially for smaller datasets[7, 32], it can result in limiting the expressiveness of the network due to restrictions on

36th Conference on Neural Information Processing Systems (NeurIPS 2022).

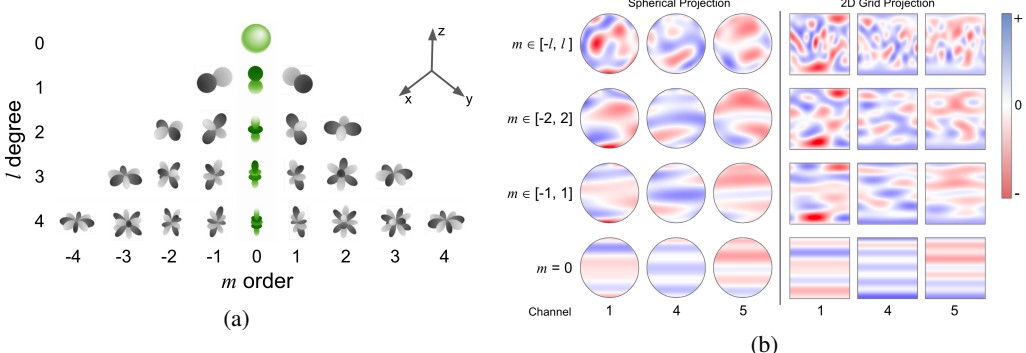

Figure 1: (a) Illustration of spherical harmonics up to $l = 4$. Note the $m = 0$ bases are symmetric about the z-axis (center green). (b) Visualization of 3 spherical channels with $l = 8$ when varying the number of orders: all, $m \in [-2, 2]$, $m \in [-1, 1]$, $m = 0$. Note how the resolution decreases with fewer $m$ until values are constant for a given z with $m = 0$. Spherical projections are shown on the left (only half of the channel is visible) and a 2D grid projection (polar and azimuthal) of the same channels is shown on the right.

non-linear transformations for equivariant models [3, 42, 34, 36], or limiting interactions to pairs [35, 38, 45], triplets [13] or quadruplets of atoms [28, 14, 15] for invariant models. Alternatively, a non-equivariant model [23] can provide more freedom to the model, but lead to the model needing to learn approximate equivariance through methods such as data augmentation [40]. To draw an analogy with image detection, the use of a CNN [27] provides translation equivariance and removes the need for the network to learn how to detect the same object at different locations. However, most CNNs are not equivariant to scale or rotation [44, 43], but are still effective in learning approximate equivariance through rotation and scale diversity in the training data. What is the analogous balance of constraints for modeling atomic systems?

In this paper, we propose a GNN [19, 47] that balances the use of model constraints to aid in generalization while providing the network with the flexibility to learn accurate representations. We introduce the Spherical Channel Network (SCN) that explicitly models relative orientations of all neighboring atoms; information that is critical to accurately predicting atomic properties. Each node's embedding is a set of functions defined on the surface of a sphere $(S^2 \to \mathbb{R})$. The functions are represented using spherical harmonics, similar to approaches that build strictly equivariant models [42, 1, 3, 10]. During message passing, angular information between atoms is conveyed by rotating or steering [5] the embeddings based on each edge's orientation [41, 28]. We identify an expanded set of spherical harmonic coefficients that are invariant to rotation, which can provide rich information while maintaining a message's rotation equivariance. In addition, we demonstrate that if the equivariance constraint is relaxed, improved performance can be achieved by using additional coefficients. We further improve the expressivity of the network by performing a pointwise non-linear function, which is only approximately equivariant, on the embeddings during message aggregation.

We demonstrate our Spherical Channel Network on the large-scale Open Catalyst 2020 (OC20) dataset [6], which contains atomic structures useful for numerous applications important to addressing climate change. State-of-the-art results are achieved for atomic force and initial structure to relaxed energy prediction with improvements of $8\% - 11\%$. In addition, we demonstrate our model is more sample efficient as compared to other state-of-the-art models.

## 2   Approach

Given an atomic structure with $n$ atoms, our goal is to predict the structure's energy $E$ and the per-atom forces $\boldsymbol{f}_i$ for each atom $i \in n$. These values are estimated using a Graph Neural Network (GNN) [19, 47] where each node represents an atom and edges represent nearby atoms. As input, the network is given the distance $d_{ij}$ between atoms $i$ and $j$, and each atom's atomic number $a_i$. The neighbors $N_i$ for an atom $i$ are determined using a fixed distance threshold, or by picking a fixed number of closest atoms.

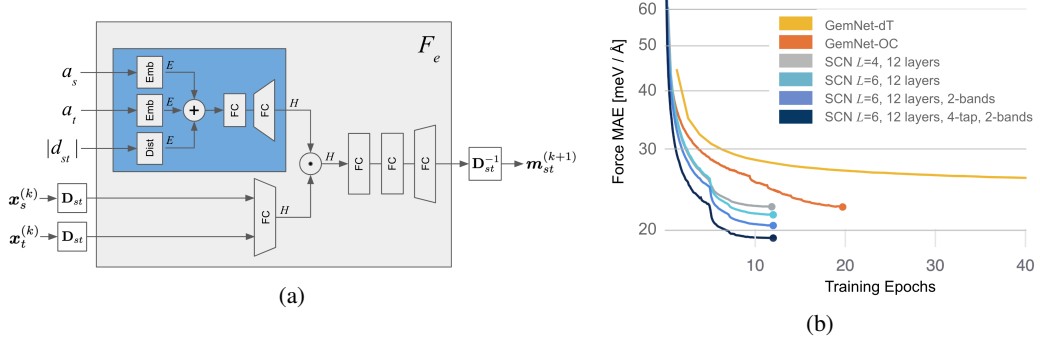

(a)

(b)

Figure 2: (a) Block diagram of message passing function $\boldsymbol{F}_e$ in Equation 2 for source atom $s$ to target atom $t$. The atomic numbers $a_s$ and $a_t$, distance between the atoms $|d_{st}|$, and embeddings $\boldsymbol{x}_s^{(k)}$ and $\boldsymbol{x}_t^{(k)}$ are given as input. (b) Training curves for SCN and GemNet models for force MAEs per epoch evaluated on a 30k subset of the validation ID dataset. Note how the SCN model is significantly more sample efficient during training.

## 2.1 Node Embeddings

The angular or relative orientation information between atoms is critical to accurately modeling atomic properties such as energies and forces [13, 14]. Inspired by this, our node embedding models the angular information from all neighboring atoms using spherical functions. Each node $i$'s embedding $s_i$ is a set of $C$ functions or channels represented on a sphere ($S^2 \rightarrow \mathbb{R}$), whose argument is a 3D unit vector indicating the orientation. That is, $s_{ic}(\hat{\boldsymbol{r}})$ is the value of channel $c$ for node $i$ for some orientation $\hat{\boldsymbol{r}} \in \mathbb{R}^3$. Since the spherical channels contain orientation information over the entire sphere, the network may reason about geometric information for all neighboring atoms and not just atom pairs, triplets, etc. Each spherical channel $c$ may be represented using several different approaches, such as discrete 2D grids sampled over the sphere, or spherical harmonics. As we describe later, we use spherical harmonics due to their property of being SO(3)-equivariant (3D rotation equivariant).

Using real spherical harmonics, a function $s_c(\hat{\boldsymbol{r}})$ defined on the sphere is represented by a set of weighted spherical harmonics $Y_{lm}$ with $s_c(\hat{\boldsymbol{r}}) = \sum_{l,m} \boldsymbol{x}_{lmc} Y_{lm}(\hat{\boldsymbol{r}})$ where $l$ and $m$ are the degree and order of the basis functions with $m \in [-l, l]$ and $l \in L$. We refer to the spherical harmonic coefficients, $\boldsymbol{x}_{lmc}$, as the coefficients of the spherical channel. For every degree $l$ there exists $2l + 1$ spherical harmonics (see Figure 1(a) for an illustration of functions for $l \leq 4$), which results in a total of $(L + 1)^2$ basis functions and coefficients up to degree $L$. Therefore for a maximum degree of $L$ with $C$ channels, $\boldsymbol{x}$ has size $(L + 1)^2 \times C$.

The spherical channels are updated by the GNN through message passing for $K$ layers to obtain the final node embeddings $S^{(K)}$. $S^{(0)}$ is initialized from an embedding based on the atom's atomic number $a_i$ for $l = 0$ coefficients and the $l \neq 0$ coefficients are set to zero. The nodes' embeddings are updated by first calculating a set of messages $m_{ij}$ for each edge, which are then aggregated at each node. Finally, the energy and forces are estimated from $S^{(K)}$. We describe each of these steps in turn.

## 2.2 Message Passing

Given a target node $t$ and its neighbors $s \in N_t$ we want to update the embeddings $\boldsymbol{x}_t^{(k)}$ at iteration $k \in K$. The embeddings $\boldsymbol{x}_t^{(k)}$ are a set of spherical harmonic coefficients indexed by their degree $l$, order $m$, and channel $c$. An important and useful property of spherical harmonics is the function represented by the coefficients is steerable [11, 5], *i.e.*, it can be rotated in 3D space using a linear transformation of the coefficients. Specifically, for a 3D rotation matrix $\mathbf{R}$ there exists a matrix called a Wigner D-matrix $\mathbf{D}^l$ of size $(2l + 1 \times 2l + 1)$ that rotates the coefficients of degree $l$ by the rotation $\mathbf{R}$. If $\boldsymbol{x}_{ilc}$ are all coefficients across orders $m$ for node $i$ of degree $l$ and channel $c$, then for any 3D

rotation $\mathbf{R}$ there exists a $\mathbf{D}^l$ that for all orientations $\hat{\boldsymbol{r}}$:

$$\boldsymbol{x}_{ilc} \cdot Y_l(\mathbf{R}\hat{\boldsymbol{r}}) = (\mathbf{D}^l \boldsymbol{x}_{ilc}) \cdot Y_l(\hat{\boldsymbol{r}}). \tag{1}$$

When calculating the message $m_{st}$ from atom $s$ to atom $t$, we want to use the information contained in both $\boldsymbol{x}_s^{(k)}$ and $\boldsymbol{x}_t^{(k)}$ given the context of the edge's orientation $\hat{\boldsymbol{d}}_{st}$, $\hat{\boldsymbol{d}}_{st} = \boldsymbol{d}_{st}/|\boldsymbol{d}_{st}|$. We do this by rotating the embeddings by $\mathbf{R}_{st}$ for which $\mathbf{R}_{st}\hat{\boldsymbol{d}}_{st} = [0,0,1]^\top$, *i.e.*, the direction of the edge's unit vector $\hat{\boldsymbol{d}}_{st}$ is aligned with the z-axis. Thus the orientation of the atoms with respect to each other is implicit in the rotated embeddings. This simplifies the task of the network since it only needs to learn the relationship between atoms that are aligned along the z-axis and not an arbitrary rotation. After calculating the messages using a neural network $\boldsymbol{F}_e$, they are rotated back to the global coordinate frame:

$$\boldsymbol{m}_{st}^{(k+1)}(\boldsymbol{x}_s^{(k)}, \boldsymbol{x}_t^{(k)}, \boldsymbol{d}_{st}, a_s, a_t) = \mathbf{D}_{st}^{-1}\boldsymbol{F}_e(\mathbf{D}_{st}\boldsymbol{x}_s^{(k)}, \mathbf{D}_{st}\boldsymbol{x}_t^{(k)}, |\boldsymbol{d}_{st}|, a_s, a_t), \tag{2}$$

where the matrices $\mathbf{D}_{st}$ and $\mathbf{D}_{st}^{-1}$ perform the rotation on the embeddings' coefficients corresponding to the 3D rotation matrices $\mathbf{R}_{st}$ and $\mathbf{R}_{st}^{-1}$ respectively. To simplify notation, we assume the coefficients in $\boldsymbol{x}_s$ across degree $l$ and order $m$ are flattened, and $\mathbf{D}_{st} \in \mathbb{R}^{(L+1)^2} \times \mathbb{R}^{(L+1)^2}$ is a block diagonal matrix constructed from the set of Wigner D-matrices across degrees, $l \in L$. $\mathbf{D}_{st}^{-1}$ is the inverse of $\mathbf{D}_{st}$, with $\mathbf{D}_{st}^{-1} = \mathbf{D}_{st}^\top$. In addition to the rotated embeddings, $\boldsymbol{F}_e$ is also provided input edge information that is invariant to rotations; $|\boldsymbol{d}_{st}|$ is the magnitude of the distance between atoms $s$ and $t$, and $a_s$ and $a_t$ their respective atomic numbers. Note that the rotation matrix $\mathbf{R}_{st}$ is not unique, since the roll rotation around the vector $\boldsymbol{d}_{st}$ is not specified and is randomly sampled during training. The implications of this are discussed below in Section 2.2.1.

The message function $\boldsymbol{F}_e$ is computed using a neural network as illustrated in Figure 2(a). The atomic numbers $a_s$ and $a_t$ are used to look up two independent embeddings of size $E = 128$, and a set of 1D basis functions are used to represent $|\boldsymbol{d}_{st}|$ using equally spaced Gaussians every 0.02 Å from 0 to the 8 Å with $\sigma = 0.04$ followed by a linear layer with $E$ outputs. Their values are added together and passed through a neural network to produce a vector of size $H$. The rotated embeddings $\mathbf{D}_{st}\boldsymbol{x}_s^{(k)}$ and $\mathbf{D}_{st}\boldsymbol{x}_t^{(k)}$ are concatenated and passed through a single layer neural network to produce another vector of size $H$. These two are multiplied together to combine the edge information with the information contained in the rotated embeddings. Two more fully connected layers are performed with each followed by a SiLU non-linear activation function [22]. Finally, a single linear layer is used to expand the output to the original size of the embeddings.

### 2.2.1 Message Equivariance

For the network to be equivariant to rotations, our message function (Equation (2)) needs to be equivariant. In general, this is not the case since the matrix $\mathbf{D}_{st}$ is not unique, *i.e.*, the roll rotation around the vector $\boldsymbol{d}_{st}$ is not specified, and in practice it is randomly chosen. The roll rotation corresponds to a rotation about the z-axis after rotating the coefficients $\boldsymbol{x}_s^{(k)}$ and $\boldsymbol{x}_t^{(k)}$ by $\mathbf{D}_{st}$. Due to this, all of the rotated coefficients will vary based on the random roll rotation chosen, except the $m = 0$ coefficients that are symmetric about the z-axis; see Appendix D and the $m = 0$ bases highlighted in green in Figure 1(a). If rotation equivariance is desired for messages, the input and output coefficients to $\boldsymbol{F}_e$ can be restricted to only those for which $m = 0$. Since the other inputs $|\boldsymbol{d}_{st}|$, $a_s$ and $a_t$ to $\boldsymbol{F}_e$ are also invariant to rotations, $\boldsymbol{F}_e$ is invariant to rotations if only $m = 0$ coefficients are used. The resulting messages $\boldsymbol{m}_{st}$ are equivariant to rotations once they are rotated back to the global coordinate frame using $\mathbf{D}_{st}^{-1}$. While equivariance is a desirable property, as we demonstrate later, using $m \in [-1, 1]$ coefficients can help improve performance even though equivariance is not strictly enforced.

If we choose to use coefficients beyond just $m = 0$ for $\boldsymbol{F}_e$ for message passing in Equation (2), two approaches may be taken. The first is to simply add the coefficients to the inputs and outputs of $\boldsymbol{F}_e$ and assume the neural network will learn a roughly equivariant mapping through the random sampling of the roll rotation. As we demonstrate in Appendix A, the learned functions are indeed roughly equivariant. The second strategy takes a more direct approach to encouraging the network to learn equivariant mappings by taking advantage of the proprieties of the coefficients as they're rotated by $\phi$ about the z-axis. If $\mathbf{D}_\phi$ rotates a set $x$ of coefficients about the z-axis by $\phi$, the $m = 0$

coefficients are constant as a function of $\phi$, while the $m \in \{-1, 1\}$ coefficients are sine and cosine functions of $\phi$ (see Appendix D):

$$(\mathbf{D}_\phi \boldsymbol{x})_{(0)} = \boldsymbol{\gamma}, \tag{3}$$

$$(\mathbf{D}_\phi \boldsymbol{x})_{(-1)} = \boldsymbol{\alpha} \sin(\phi + \boldsymbol{\beta}) = (\mathbf{D}_{(\phi - \frac{\pi}{2})} \boldsymbol{x})_{(1)}, \tag{4}$$

$$(\mathbf{D}_\phi \boldsymbol{x})_{(1)} = \boldsymbol{\alpha} \cos(\phi + \boldsymbol{\beta}) = (\mathbf{D}_{(\phi + \frac{\pi}{2})} \boldsymbol{x})_{(-1)}, \tag{5}$$

for some set of vectors $\boldsymbol{\alpha}$, $\boldsymbol{\beta}$ and $\boldsymbol{\gamma}$ of size $L$. $\boldsymbol{x}_{(0)}$ are the $m = 0$ coefficients and similarly for $\boldsymbol{x}_{(-1)}$ and $\boldsymbol{x}_{(1)}$ for the $m = -1$ and $m = 1$ coefficients respectively. Similar proprieties hold for $m > 1$ and $m < -1$. We take advantage of these properties to encourage the output of the message block towards equivariance by computing $\boldsymbol{F}_e$ at multiple rotations $\phi$ about the z-axis:

$$\boldsymbol{F}_e^\phi = \boldsymbol{F}_e(\mathbf{D}_\phi \mathbf{D}_{st} \boldsymbol{x}_s^{(k)}, \mathbf{D}_\phi \mathbf{D}_{st} \boldsymbol{x}_t^{(k)}, |\boldsymbol{d}_{st}|, a_s, a_t), \tag{6}$$

For $m \in [-1, 1]$, we compute four samples or "taps" at $\phi \in \{0, \frac{1}{2}\pi, \pi, \frac{3}{2}\pi\}$, and combine them based on $m$ using:

$$
\begin{aligned}
\boldsymbol{m}_{st}^{(k+1)} = \quad & \mathbf{D}_{st}^{-1} \frac{1}{4} \left( \boldsymbol{F}_{e(0)}^0 + \boldsymbol{F}_{e(0)}^{\frac{1}{2}\pi} + \boldsymbol{F}_{e(0)}^\pi + \boldsymbol{F}_{e(0)}^{\frac{3}{2}\pi} \right) + \\
& \mathbf{D}_{st}^{-1} \frac{1}{4} \left( \boldsymbol{F}_{e(-1)}^0 - \boldsymbol{F}_{e(1)}^{\frac{1}{2}\pi} - \boldsymbol{F}_{e(-1)}^\pi + \boldsymbol{F}_{e(1)}^{\frac{3}{2}\pi} \right) + \\
& \mathbf{D}_{st}^{-1} \frac{1}{4} \left( \boldsymbol{F}_{e(1)}^0 + \boldsymbol{F}_{e(-1)}^{\frac{1}{2}\pi} - \boldsymbol{F}_{e(1)}^\pi - \boldsymbol{F}_{e(-1)}^{\frac{3}{2}\pi} \right)
\end{aligned} \tag{7}
$$

where $\boldsymbol{F}_{e(0)}^\phi$, $\boldsymbol{F}_{e(-1)}^\phi$ and $\boldsymbol{F}_{e(1)}^\phi$ are the output coefficients for $m = 0$, $m = -1$ and $m = 1$ respectively. The top line of Equation 7 is simply taking the average, since the $m = 0$ coefficients should be constant regardless of $\phi$. Similarly for $m \in \{-1, 1\}$, the second and third lines average 4 values that should be equal if Equations 4 and 5 hold, e.g., $\boldsymbol{\alpha} \sin(\phi + \boldsymbol{\beta}) = -\boldsymbol{\alpha} \cos(\phi + \boldsymbol{\beta} + \pi/2)$. If higher $m$ are desired, similar calculations can be performed. However, a larger number of taps will be needed, e.g., $m \in [-2, 2]$ requires 8 taps separated by $\frac{1}{4}\pi$ radians.

The number of coefficients used during message passing when calculating $\boldsymbol{F}_e$ may be varied based on the edge properties to reduce the memory required by the network. For instance, atoms that are far away from each other may not need the same resolution of the spherical functions as those close together, so a lower maximum degree $L$ may be used. By training several message passing networks with varying degrees for different ranges of edge distances, memory usage can be reduced while not impacting accuracy. Similarly, the range of $m$ can typically be truncated at $[-1, 1]$ or $[-2, 2]$ since their observed utility to the network is substantially reduced for values of $m$ greater than 2 (or less than -2). See Figure 1(b) for examples of the spherical channels when the range of $m$ is reduced. Note that while the number of coefficients used by $\boldsymbol{F}_e$ may be reduced, the node embeddings maintain their original size based on the maximum $L$, so they may be able to aggregate information from all neighboring atoms at different distances and angles.

## 2.3  Message Aggregation

At this point in the network, the per edge messages are computed and rotated back to the global coordinate frame. Ideally, to allow for complex interactions between the messages, a non-linear function would be applied after summing all messages directed at a target node. If we did this by naively passing the summed coefficients $\boldsymbol{m}_t^{(k+1)} = \sum_s \boldsymbol{m}_{st}^{(k+1)}$ through a fully connected neural network, the network may have difficulties learning representations that are approximately rotation equivariant (a fundamental property of atom forces). Another option is to place constraints on the non-linearities performed to enforce equivariance [26, 5, 36, 42, 3, 2].

We propose applying an unconstrained non-linear pointwise function on the sphere, i.e., a function that is applied at every orientation without knowledge of the orientation from which the point was sampled [9]. For our function, we use a non-linear neural network $\boldsymbol{F}_c$ ($\mathbb{R}^C \to \mathbb{R}^C$) that combines information across channels. In practice, $\boldsymbol{F}_c$ is applied at a discrete number of orientations, which

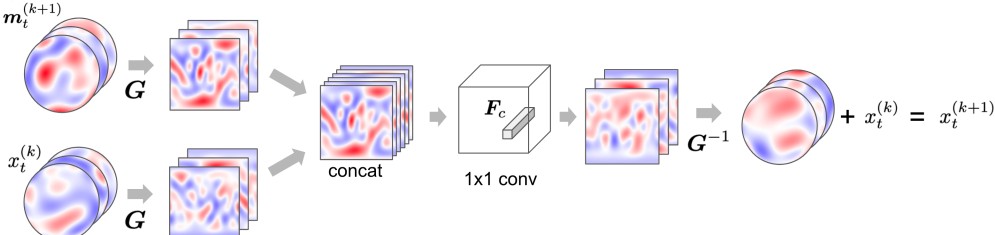

Figure 3: Illustration of message aggregation. The summed messages $m_t^{(k+1)} = \sum_s m_{st}^{(k+1)}$ and previous embedding $x_t^{(k)}$ are converted from a spherical harmonic representation to a spherical grid representation using $G$. The channels are concatenated and passed through a 3 layer $1 \times 1$ CNN. Each layer is followed by a SiLU activation. Finally, the channels are converted back to a spherical harmonic representation using $G^{-1}$ and added to $x_t^{(k)}$ to create $x_t^{(k+1)}$.

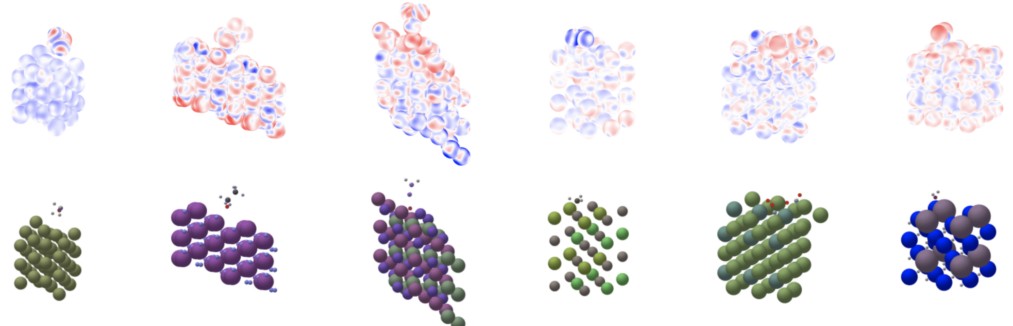

Figure 4: Illustration of different spherical channels for six structures (top). Blue indicates a positive value and red a negative value on the sphere. The spherical channels were sampled from the last layer before the output blocks using a model with 12 layers and $L = 6$. Note, how some channels have a higher activation for adsorbates (darker colors), certain elements or the bottom or top of the surface. Illustrations of the structures are shown on the bottom with different colors indicating different elements.

results in a transformation that we numerically demonstrate (see Appendix A) is approximately equivariant to rotations:

$$x_t^{(k+1)} = x_t^{(k)} + G^{-1}\left(F_c\left(G(m_t^{(k+1)}), G(x_t^{(k)})\right)\right), \tag{8}$$

where $G$ is a function that converts a spherical function represented by spherical harmonic coefficients, to one represented by point samples on a sphere. For $G$ we use a spherical coordinate system (polar and azimuthal) to generate a 2D discrete representation of the spherical functions, see Figure 3. A $1 \times 1$ convolutional neural network $F_c$ is applied to each discrete sample containing $C$ channels, and the result is then converted back to the spherical harmonic representation using $G^{-1}$. This is analogous to having a 2D image represented in the frequency domain, converting it to the spatial domain using an inverse discrete Fourier transform, applying a transformation and converting back to the frequency domain. Since the same operation is applied to all orientations on the sphere, a transformation that is approximately equivariant (up to discrete sampling limitations) to rotations may be learned, see Appendix A for a more detailed discussion.

A diagram of the network is shown in Figure 3. To provide additional information to the network, $x_t^{(k)}$ is also converted to a spherical grid representation, concatenated with $G(m_t^{(k+1)})$ and provided as input to the neural network to compute the final updates to the coefficients $x_t^{(k+1)}$ at iteration $k+1$. The network can learn circular functions on the sphere if varying resolutions of $x^{(k)}$ are provided to the network, similar to Difference of Gaussian filters for 2D images [30]. The resolution of $x^{(k)}$ can be reduced by lowering the degree $L$ before transforming by $G$. In our experiments, if multiple resolutions or bands are used, we use resolutions of degree $L$ and $L-1$. Three layers of $1 \times 1$ convolutions on $2C$ channels (or $4C$ if two bands are used) are performed with each followed by a

SiLU non-linear activation function. To avoid aliasing, the spherical grid is sampled at a resolution of $2 * (L + 1)$. See Figure 4 for several example illustrations of spherical channels for different atomic structures.

## 2.4 Energy and Forces Estimation

We compute the energy $E$ by first estimating a per-atom energy using a pointwise function on the sphere and taking its integral over all possible orientations. The per-atom energy estimates are summed to obtain the overall system's energy:

$$E = \sum_i \int \boldsymbol{F}_{energy} \left( s_i^{(K)}(\hat{\boldsymbol{r}}) \right) d\hat{\boldsymbol{r}}, \tag{9}$$

where $\boldsymbol{F}_{energy}$ is a three layer fully connected neural network ($\mathbb{R}^C \to \mathbb{R}^1$) with SiLU activation functions.

Forces may be calculated using two approaches: First, by calculating the gradients of the energy with respect to the atom positions. This approach enforces energy conservation, but due to the need to back-propagate the gradients is computationally much more expensive. The second approach computes the forces in a manner similar to energy prediction, which is computationally more efficient, but does not enforce energy conservation. In this approach, the forces are calculated by estimating a force magnitude $|\boldsymbol{f}| = \boldsymbol{F}_{force}(s_i^{(K)}(\hat{\boldsymbol{r}}))$ in every direction $\hat{\boldsymbol{r}}$ over the sphere. Integration is performed after multiplying the magnitude by the orientation $\hat{\boldsymbol{r}}$ to obtain directional vectors:

$$\boldsymbol{f}_i = \int \hat{\boldsymbol{r}} \boldsymbol{F}_{force} \left( s_i^{(K)}(\hat{\boldsymbol{r}}) \right) d\hat{\boldsymbol{r}}, \tag{10}$$

where $\boldsymbol{F}_{force}$ is a three layer fully connected neural network ($\mathbb{R}^C \to \mathbb{R}^1$). In practice, a discrete approximation of the integral is performed for equations (9) and (10) using a set of 128 evenly distributed points on the sphere (see Appendix C.2). Note that the application of a pointwise function of an arbitrary neural network at a finite number of discrete orientations is not invariant (energies) or equivariant (forces) to rotations. However in practice, numerically this operation does closely approximate these properties (see Appendix A.2).

## 3 Experiments

We present results on the Open Catalyst 2020 (OC20) dataset [6] that is released under a Creative Commons Attribution 4.0 License. OC20 contains over 130M training examples for the task of predicting atomic energies and forces for catalysts used in renewable energy storage and other important applications [48]. This dataset is a popular benchmark for the ML community. We begin by comparing results across all tasks on the test set. Next, we show numerous ablation studies comparing model variations on the smaller OC20 2M dataset. Finally, since several papers only report results on the IS2RE task using approaches that directly predict relaxed energies, we also train a model specifically for this task and approach, and compare in Appendix B.

### 3.1 Implementation Details

The implementation of spherical harmonics and their transformations uses the code provided by Euclidean neural networks (e3nn) [16]. For message passing, two resolutions of spherical harmonics are used based on a atom's nearest neighbors, see Appendix C.1 for exact parameters. Unless otherwise stated $C = 128$, $K = 16$, $H = 1024$, $E = 128$, and for $\boldsymbol{F}_e$ in message passing only orders $m \in [-1, 1]$ are used. All forces are estimated directly as an output of the network, unless stated that the energy conserving gradient-based approach was used. During training, the coefficients for the force and energy losses are 100 and 2 respectively. Training is performed using the AdamW optimizer [29] with a learning rate of 0.0004. The effective batch size is increased using data parallelism and PyTorch's Automatic Mixed Precision (AMP). All model code will be open sourced with an MIT license in the Open Catalyst Github repo. Please see the supplementary for details on the PaiNN[36] baseline.

| Model | $L$ | # layers | $H$ | # batch | Samples / GPU sec. | S2EF Energy MAE [meV] ↓ | Force MAE [meV/Å] ↓ | Force Cos ↑ | EFwT [%] ↑ | IS2RE Energy MAE [meV] ↓ | EwT [%] ↑ |
|---|---|---|---|---|---|---|---|---|---|---|---|
| Median | | | | | | | | | | | |
| SchNet [38] | | | | | | 1400 | 78.3 | 0.109 | 0.00 | - | - |
| DimeNet++ [12] | | | | | | 805 | 65.7 | 0.217 | 0.01 | - | - |
| SpinConv [41] | | | | | | 406 | 36.2 | 0.479 | 0.13 | - | - |
| GemNet-dT [14] | | | | | 25.8 | 358 | 29.5 | 0.557 | 0.61 | 438 | - |
| GemNet-OC [15] | | | | | 18.3 | 286 | 25.7 | 0.598 | 1.06 | 407 | - |
| SCN No rotation | 4 | 12 | 512 | 64 | 8.5 | 410 | 67.7 | 0.192 | 0.01 | - | - |
| SCN No 1x1 conv | 6 | 12 | 1024 | 128 | 9.1 | 313 | 26.2 | 0.579 | 0.74 | - | - |
| SCN $m = 0$ | 6 | 12 | 1024 | 128 | 10.4 | 307 | 26.5 | 0.588 | 0.83 | - | - |
| SCN $m \in [-1, 1]$ | 6 | 12 | 1024 | 128 | 8.3 | 302 | 24.6 | 0.601 | 0.93 | - | - |
| SCN $m \in [-2, 2]$ | 6 | 12 | 1024 | 96 | 6.9 | 301 | 23.4 | 0.612 | 1.01 | - | - |
| SCN $m \in [-l, l]$ | 4 | 12 | 512 | 64 | 7.6 | 297 | 24.6 | 0.595 | 0.92 | - | - |
| SCN grad-forces | 4 | 12 | 512 | 128 | 1.2 | 307 | 26.2 | 0.573 | 0.81 | - | - |
| SCN direct-forces | 4 | 12 | 512 | 128 | 12.1 | 303 | 25.3 | 0.592 | 0.87 | - | - |
| SCN | 2 | 12 | 256 | 128 | 12.9 | 312 | 27.6 | 0.568 | 0.70 | - | - |
| SCN | 4 | 12 | 512 | 128 | 12.1 | 303 | 25.3 | 0.592 | 0.87 | - | - |
| SCN | 6 | 12 | 1024 | 128 | 8.3 | 302 | 24.6 | 0.601 | 0.93 | - | - |
| SCN | 8 | 12 | 1024 | 64 | 5.9 | 300 | 23.2 | 0.620 | 1.15 | - | - |
| SCN | 6 | 12 | 1024 | 64 | 7.7 | 299 | 24.3 | 0.605 | 0.98 | - | - |
| SCN 2-band | 6 | 12 | 1024 | 64 | 5.1 | 292 | 23.1 | 0.622 | 1.18 | - | - |
| SCN 4-tap | 6 | 12 | 1024 | 64 | 3.7 | 296 | 22.7 | 0.638 | 1.27 | - | - |
| SCN 4-tap 2-band | 6 | 12 | 1024 | 64 | 3.5 | **279** | **22.2** | 0.643 | **1.41** | 371 | 11.0 |
| SCN $m = 0$ | 6 | 16 | 1024 | 96 | 7.4 | 300 | 25.7 | 0.600 | 0.96 | 394 | 9.6 |
| SCN $m = 0$ | 8 | 16 | 1024 | 64 | 4.8 | 296 | 25.3 | 0.608 | 1.01 | 389 | 9.9 |
| SCN | 6 | 16 | 1024 | 96 | 5.9 | 287 | 22.8 | 0.623 | 1.22 | 371 | 10.5 |
| SCN | 8 | 16 | 1024 | 96 | 3.5 | **283** | 22.7 | 0.627 | 1.22 | **364** | **11.3** |
| SCN 4-tap | 6 | 16 | 1024 | 64 | 2.6 | **282** | 22.2 | **0.648** | 1.37 | 378 | 10.7 |
| SCN 4-tap 2-band | 6 | 16 | 1024 | 64 | 2.3 | **279** | **21.9** | **0.650** | **1.46** | 373 | **11.0** |

Table 1: Results on the OC20 2M training dataset and ablation studies for SCN model variations. The validation results are averaged across the four OC20 Validation set splits. All SCN models are trained on 16 GPUs for 12 epochs with the learning rate reduced by 0.3 at 5, 7, 9, and 11 epochs, except SCN with $L = 8$ and 16 layers that used 32 GPUs to obtain a larger batch size. Batch sizes vary based on the number of instances that can be fit in 32GB RAM.

## 3.2 OC20 Tasks

We compare against numerous models trained on the OC20 2M, All and MD datasets across the Structure to Energy and Forces (S2EF), Initial Structure to Relaxed Structure (IS2RS) and Initial Structure to Relaxed Energy (IS2RE) tasks [6]. OC20 2M, OC20 All and OC20 MD have 2, 133, and 38 million training examples respectively. The 12 and 16 layer SCN models significantly outperform state-of-the-art GemNet-OC [15] on S2EF force prediction ($\approx 14\%$ improvement) and IS2RE energy predictions ($\approx 10\%$ improvement) when trained on OC20 2M, Table 1. Comparable results are found for energy MAE on S2EF with GemNet-OC. We also report throughput efficiency. Note that while the SCN models process fewer samples per second, they are more sample efficient than GemNet (Figure 2(b)). Efficiency improvements have also been noted for fully equivariant models [3].

We compare our largest SCN models trained on the All and the All+MD datasets (133M + 38M examples) across all OC20 tasks in Table 2. For these experiments we use a deep 1-tap model, and a slightly shallower 4-tap model since the use of 4-taps requires more memory. Our model achieves state-of-the-art results for force MAE ($\approx 9\%$ improvement) and force cosine for the S2EF tasks. The SCN $L = 6$ model performs similarly to the recently released GemNet-OC model [15] on energy MAE for S2EF. On the challenging relaxation based IS2RE task that requires models to predict both accurate forces and energies, SCN outperforms GemNet-OC by over 7%. Note that we only train a single model to predict both energy and forces, where GemNet-OC-L-F+E + MD is the combination of two models, one trained specifically for forces and the other for energy.

## 3.3 Ablation studies

We explore numerous model variations in Table 1 trained on the OC20 2M dataset and evaluated on the OC20 validation set. The first set of ablation experiments explores reducing the complexity of our model. If messages are not rotated before and after applying $\boldsymbol{F}_m$ in Equation 2, *i.e.*, $\mathbf{A}_{st}$

| Model | #Params | Train time | S2EF | | | | IS2RS | | IS2RE |
|---|---|---|---|---|---|---|---|---|---|
| | | | Energy MAE meV ↓ | Force MAE [meV/Å] ↓ | Force Cos ↑ | EFwT [%] ↑ | AFbT [%] ↑ | ADwT [%] ↑ | Energy MAE meV ↓ |
| Median | – | | 2258 | 84.4 | 0.016 | 0.01 | - | - | - |
| | | | | | | **Train OC20 All** | | | |
| SchNet [38, 6] | 9.1M | 194d | 540 | 54.7 | 0.302 | 0.00 | - | 14.4 | 764 |
| PaiNN [36] | 20.1M | 67d | 341 | 33.1 | 0.491 | 0.46 | 11.7 | 48.5 | 471 |
| DimeNet++-L-F+E [12, 6] | 10.7M | 1600d | 480 | 31.3 | 0.544 | 0.00 | 21.7 | 51.7 | 559 |
| SpinConv (direct-forces) [41] | 8.5M | 275d | 336 | 29.7 | 0.539 | 0.45 | 16.7 | 53.6 | 437 |
| GemNet-dT [14] | 32M | 492d | 292 | 24.2 | 0.616 | 1.20 | 27.6 | 58.7 | 400 |
| GemNet-OC [15] | 39M | 336d | **233** | 20.7 | 0.666 | 2.50 | 35.3 | 60.3 | 355 |
| SCN $L$=8 $K$=20 | 271M | 645d | 244 | **17.7** | **0.687** | 2.59 | **40.3** | **67.1** | **330** |
| | | | | | | **Train OC20 All + MD** | | | |
| GemNet-OC-L-E [15] | 56M | 640d | **230** | 21.0 | 0.665 | 2.80 | - | - | - |
| GemNet-OC-L-F [15] | 216M | 765d | 241 | 19.0 | 0.691 | **2.97** | 40.6 | 60.4 | - |
| GemNet-OC-L-F+E [15] | - | - | - | - | - | - | - | - | 348 |
| SCN $L$=6 $K$=16 4-tap 2-band | 168M | 414d | **228** | 17.8 | **0.696** | 2.95 | **43.3** | 64.9 | 328 |
| SCN $L$=8 $K$=20 | 271M | 1280d | 237 | **17.2** | **0.698** | 2.89 | **43.6** | **67.5** | **321** |

Table 2: Comparison of SCN to existing GNN models on the S2EF, IS2RS and IS2RE tasks when trained on the All or All+MD datasets. Average results across all four test splits are reported. We mark as bold the best performance and close ones, *i.e.*, within 0.5 meV/Å MAE, which we found to empirically provide a meaningful performance difference. Training time is in GPU days. Median represents the trivial baseline of always predicting the median training energy or force across all the validation atoms. The SCN $L = 8$ model has $K = 20$ layers, $C = 128$ and $E = 256$, while the SCN $L = 6$ model has $K = 16$ layers, $C = 128$, $E = 128$ and an energy loss coefficient of 4.

is an identity matrix, the results are significantly worse for energies and even more so for forces since forces are highly dependent on angular information. If we replace the non-linear aggregation (Equation 8) with a simple summation (No 1x1 conv) in a model with $m \in [-1, 1]$, the results also degrade across all metrics. Results improve for $m \in [-1, 1]$ as compared to only using $m = 0$ coefficients, which demonstrates that if the equivariant to rotation constraint is relaxed, improved results may be achieved. For higher $m$, $m \in [-2, 2]$ or $m \in [-4, 4]$ diminishing returns are noticed with increased computational cost.

In some applications, such as molecular dynamics, energy conserving models are needed. We trained an energy conserving model that estimates forces based on the gradients of the energy with respect to the atom positions. Since this approach is more expensive both in memory and computation, we used a smaller model with $L = 4$, $H = 512$ and 12 layers (grad-forces). While the accuracy of the model is good, the results are slightly worse and is 10 times slower than a comparable model that directly estimates forces (direct-forces).

When scaling the network, increasing $L$ has a more significant impact on force estimation, while depth improves energy estimation. This may be due to higher $L$ resulting in greater angular resolution, which is important to force estimation. Greater depth, which allows information to travel further, may lead to better energy predictions since energy is function of the entire atomic structure.

Sampling multiple z-axis rotations when computing $\boldsymbol{F}_e$ with 4-taps (Equation 7) produces significantly improved force predictions, while using two bands ($L$ and $L-1$) when aggregating messages improves both energy and force prediction. The use of 2 bands reduces model throughput a small amount, while sampling with 4-taps reduces throughput by over $2\times$. However, sample efficiency is further increased with 4-taps, resulting in similar training times for the same accuracy, Figure 2(b).

# 4 Related work

Molecular and atomic property prediction has made significant recent progress. Models relying on hand-crafted representations like Behler-Parrinello[4] and sGDML[8] have recently been surpassed by learned feature representations using GNNs [3, 12, 14, 36]. Early GNN developments focused only on invariant representations. CGCNN[45] and SchNet[35] make energy and force estimations using only atomic species and pair-wise distance information. DimeNet[12, 13], SphereNet[28], and GemNet[14, 15] extend this to explicitly capture triplet and quadruplet angles, which are scalars. Utilizing invariant representations offers flexibility in model architecture design since the node or edge features are inherently invariant and model constraints are not needed to maintain equivariance. The

enumeration of triplets and quadruplets of atoms to calculate functions of relative or dihedral angles requires careful model design to maintain efficiency. More recently, equivariant models, capturing both scalar and equivariant features, have outperformed traditional GNNs on small molecular datasets including MD17[7] and QM9[32]. While not strictly equivariant, our work falls into this later class of GNNs for atomic property predictions.

Models that strictly enforce equivariance to the SO(3) (3D rotations) or E(3) group (3D rigid transformations) share similarities to our use of spherical channels [36, 1, 3, 9, 5]. Both represent atoms by higher-order tensors and not just scalar information. For instance, Tensor Field Networks [42], Cormorant [1], NequIP [3], SEGNN [5] and BOTNet [2] use the same spherical harmonic representations as SCNs, while PaiNN [36] is an $l = 1$ variation. All models place constraints on the operations that can be performed to ensure equivarance, such as restricting non-linear operators to invariant scalar inputs, e.g., pair-wise atom distances. In networks like TFN, geometric information of higher degree $l$ are mixed into the $l = 0$ features via tensor products on which a gated non-linear function is applied. Similar to our approach, the recently proposed SEGNN uses the spherical harmonic representation to enable steerable features based on edge orientations. SEGNN introduces steerable MLPs that enforce equivariance by restricting the learnable parameters to those that scale the Clebsch-Gordan tensor products followed by gated non-linearities [43]. In our approach, we increase the expressivity of the model by identifying a set of $m = 0$ coefficients across degrees $l$ that are invariant to rotations given an edge's orientation to which unconstrained functions may be applied. We also demonstrate that improved performance can be achieved by relaxing the requirement for strict message equivariance by including a broader set of coefficients beyond $m = 0$. Furthermore, we project the spherical channels onto a grid and perform a pointwise $1 \times 1$ convolution followed by a non-linearity. Although not strictly equivariant, the pointwise operation allows for complex mixing of different degrees of spherical harmonics resulting in rich geometric descriptions. Perhaps as a result of the network's increased expessivity, we see an increase in accuracy with higher degrees ($L = 4, 6, 8$) unlike previous approaches that typically use $L = 1$ or $L = 2$.

## 5 Discussion

A limitation of SCN is that it is very computationally expensive if energy conservation is enforced through the estimation of forces using energy gradients with respect to atom positions. This may limit its use in chemistry applications such as molecular dynamics. Non-energy conserving models are still practically useful for applications such as structure relaxations and transition state searches. The SCN model scales $O(n^2)$ with respect to $L$, which may limit using $L$ larger than 8. Using a larger set of $m$ coefficients during message passing also faces challenges, since the network may find it more difficult to learn approximately equivariant mappings. In Table 1, we saw that increased depth improves results, especially to energy prediction. It remains an open question whether deeper networks will see further improvement.

Given SCN's increased expressivity due to the relaxation of the equivariance constraint and larger model sizes, it is prone to overfitting on smaller datasets. SCN's results on datasets such as MD17[7] and QM9[32] are not state-of-the-art. However, in future work it would be interesting to explore pre-training on larger datasets, such as OC20, and see if improved results can be achieved on smaller datasets through finetuning [25].

While this work is motivated by problems we face in addressing climate change, advances in chemistry may have numerous use cases. Some of these use cases may have unintended consequences, such as the development of the Haber-Bosch process for ammonia fertilizer production that enabled the world to feed a growing population, but over fertilization has recently led to ocean "dead zones" and its production is very carbon intensive. More alarmingly, the knowledge gained from fertilizer production was used to create explosives during wartime [20]. We hope to steer research in this area towards beneficial applications by utilizing datasets such as OC20.

In conclusion, we propose a GNN that uses spherical harmonics to represent channel embeddings that contain explicit orientation information. While the network is encouraged to learn output mappings that are equivariant to rotations, best performance is achieved by relaxing the equivariant constraints and allowing for more expressive non-linear transformations. We demonstrate state-of-the-art results across numerous tasks on the large-scale OC20 dataset for modeling catalyst for applications addressing climate change.

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
