# Appendix: Table of contents

## A  Empirical measurement of equivariance to rotations

The SCN is not strictly equivariant to rotations, but depending on the design choices approximate equivarance may be achieved. We begin by empirically measuring the network's invariance and equivariance to rotation for energy and forces respectively. We accomplish this by computing the absolute difference between pairs of results computed from an input and a randomly rotated version of the input:

$$|SCN_{energy}(d, a) - SCN_{energy}(\mathbf{R}d, a)|, \tag{11}$$

$$|SCN_{force}(d, a) - \mathbf{R}^{-1} SCN_{force}(\mathbf{R}d, a)|, \tag{12}$$

where $d$ is the 3D positional difference between pairs of atoms, $a$ are their atomic numbers, and $\mathbf{R}$ is a random 3D rotation matrix. $SCN_{energy}$ and $SCN_{force}$ are the network's predictions for energy and forces respectively.

Mean Absolute Difference (MAD) results for various model choices are shown in Table 3 for models with 12 layers and $L = 6$. Differences are averaged over a model's outputs for a random 1,000 atomic structures. There are four sources that may lead the network to predict different values for rotated versions of the input: 1) the use of $m \neq 0$ coefficients during message passing, 2) non-linear message aggregation, Equation (3), 3) the energy and force output blocks, Equations (4,5), and 4) limits to numerical precision especially when using Automatic Mixed Precision (AMP).

In Table 3, we observe that the MAD increases when $m \in [-1, 1]$ coefficients are used during message passing as compared to the $m = 0$ message passing that is equivariant. However, the MAD of the $m \in [-1, 1]$ model that uses 4 taps during message passing is nearly identical to $m = 0$ when not using AMP. This shows that the 4-tap model is close to producing equivariant results. Note MAD is non-zero even for $m = 0$ due to the other sources of errors. The $1 \times 1$ convolution results in greater rotational differences for forces, while $m \in [-1, 1]$ during message passing has a bigger impact on energies. If AMP is used, higher MADs are found due to the limits of numerical precision.

### A.1  Message aggregation

Next, we discuss the degree to which our message aggregation Equation (3) is empirically equivariant. If Equation (3) was equivariant, the following would hold:

$$\boldsymbol{G}^{-1}\left(\boldsymbol{F}_c\left(\boldsymbol{G}(\mathbf{D}\boldsymbol{m}_t^{(k+1)}), \boldsymbol{G}(\mathbf{D}\boldsymbol{x}_t^{(k)})\right)\right) = \mathbf{D}\boldsymbol{G}^{-1}\left(\boldsymbol{F}_c\left(\boldsymbol{G}(\boldsymbol{m}_t^{(k+1)}), \boldsymbol{G}(\boldsymbol{x}_t^{(k)})\right)\right) \tag{13}$$

where $\mathbf{D}$ is a block diagonal matrix containing Wigner-D matrices that perform a rotation of the coefficients. That is, the same result should be achieved whether the input or output is rotated. Since the transformation $G$ uses a discrete sampling of the sphere, Equation (3) is clearly not strictly equivariant. The neural network $F_c$ may introduce non-linearities that result in frequencies higher than those able to be represented by the maximum degree $L$ used by the spherical harmonics. As a result, the sampling used by $G$ may be below the Nyquist rate needed to avoid aliasing. This may be minimized but not eliminated by utilizing smooth activation functions such as SiLU. However, in practice we find that the use of Equation (3) with sampling resolution of $2(L + 1)$ or higher does in practice lead to results that are approximately equivariant. In Table 4, we show several empirical results with different depths and activation functions. If only a linear layer is used, the function is equivariant and there is no difference in computed values. For non-linear activations, such as ReLU and SiLU, differences of $4\% - 7\%$ are seen between the rotated versions. Percentages are the Mean

| | | Energy | | | |
|---|---|---|---|---|---|
| | | | No AMP | | AMP |
| **Model** | | MAE [meV] ↓ | MAD [meV] ↓ | % Error ↓ | MAD [meV] ↓ | % Error ↓ |
| SCN | $m = 0$ | 307 | 7.9 | 2.6% | 12.4 | 4.0% |
| SCN | $m \in [-1, 1]$ | 302 | 39.1 | 12.9% | 42.8 | 14.2% |
| SCN | $m \in [-1, 1]$, No 1x1 conv | 313 | 30.3 | 9.7% | 37.3 | 11.9% |
| SCN | $m \in [-1, 1]$, 4-tap | 294 | 6.9 | 2.3% | 19.4 | 6.6% |
| | | Force | | | |
| | | | No AMP | | AMP |
| **Model** | | MAE [meV/Å] ↓ | MAD [meV/Å] ↓ | % Error ↓ | MAD [meV/Å] ↓ | % Error ↓ |
| SCN | $m = 0$ | 26.5 | 0.4 | 1.5% | 0.9 | 3.6% |
| SCN | $m \in [-1, 1]$ | 24.6 | 3.1 | 12.6% | 4.4 | 17.9% |
| SCN | $m \in [-1, 1]$, No 1x1 conv | 26.2 | 0.5 | 2.0% | 0.6 | 2.2% |
| SCN | $m \in [-1, 1]$, 4-tap | 23.1 | 0.4 | 1.7% | 2.6 | 11.3% |

Table 3: Mean Absolute Difference (MAD) between pairs of results computed from an input and a randomly rotated version of the same input for energy and forces. The inverse rotation is performed on the second set of forces before comparing results. Mean Absolute Errors (MAE) with respect to ground truth results are shown for reference. Results are shown when using AMP and not using AMP (single precision default). The MAD values as a percentage of the total error is also provided. All values are averaged over 1,000 atomic structures using a model with 12 layers and $L = 6$. If a model is equivariant to rotations, the MAD values will be zero.

| **Model** | **Activation** | **% MAD** |
|---|---|---|
| SCN 1-Layer | Linear | 0.0% |
| SCN 1-Layer | SiLU | 4.6% |
| SCN 2-Layer | SiLU | 6.7% |
| SCN 1-Layer | ReLU | 4.2% |
| SCN 2-Layer | ReLU | 6.8% |

Table 4: The percentage difference between outputs of message aggregation when rotated. Differences are computed between pairs of examples. The first example is not rotated, the second example's inputs are rotated and the outputs rotated by its inverse, Equation (14). The percentage of the Mean Absolute Differences with respect to the mean absolute values of the outputs are shown. Results are shown for different activation functions and depths.

Absolute Difference (MAD) divided by the mean absolute value of the outputs. MAD is computed using:

$$\left| \boldsymbol{G}^{-1} \left( \boldsymbol{F}_c \left( \boldsymbol{G}(\boldsymbol{m}_t^{(k+1)}), \boldsymbol{G}(\boldsymbol{x}_t^{(k)}) \right) \right) - \mathbf{D}^{-1} \boldsymbol{G}^{-1} \left( \boldsymbol{F}_c \left( \boldsymbol{G}(\mathbf{D}\boldsymbol{m}_t^{(k+1)}), \boldsymbol{G}(\mathbf{D}\boldsymbol{x}_t^{(k)}) \right) \right) \right| \quad (14)$$

Minimal differences are seen between using SiLU or ReLU. However, higher percentage differences are observed when using two layers of non-linearities when compared to one. We use two-layers followed by a linear layer in the models for this paper.

## A.2 Energy and Force Output Blocks

Our final set of experiments measure the empirical invariance and equivariance to rotations for the output energy and force blocks. The differences between the not rotated and rotated versions is computed using:

$$\left| \sum_i \int_{\hat{\boldsymbol{r}}} \boldsymbol{F}_{energy} \left( s_i^{(K)}(\hat{\boldsymbol{r}}) \right) - \sum_i \int_{\hat{\boldsymbol{r}}} \boldsymbol{F}_{energy} \left( \mathbf{D} s_i^{(K)}(\hat{\boldsymbol{r}}) \right) \right|, \quad (15)$$

$$\left| \sum_i \int_{\hat{\boldsymbol{r}}} \hat{\boldsymbol{r}} \boldsymbol{F}_{force} \left( s_i^{(K)}(\hat{\boldsymbol{r}}) \right) - \mathbf{R}^{-1} \sum_i \int_{\hat{\boldsymbol{r}}} \hat{\boldsymbol{r}} \boldsymbol{F}_{force} \left( \mathbf{D} s_i^{(K)}(\hat{\boldsymbol{r}}) \right) \right|, \quad (16)$$

where $\mathbf{R}$ is a random 3D rotation matrix and $\mathbf{D}$ its corresponding block diagonal matrix containing the Wigner-D matrices. Results are shown in Table 5. For both energy and forces the results are nearly equivariant, and demonstrate the outputs blocks have negligible negative impact on the overall network's equivariance. This is likely due to the force and energy output blocks integrating over the entire sphere, which may result in any differences negating each other.

| Model | Activation | Energy | | Force | |
|---|---|---|---|---|---|
| | | MAE [meV] | MAD [meV] | MAE Force [meV/Å] | MAD [meV/Å] |
| SCN 3-Layer | SiLU | 302 | 0.95 | 24.6 | 0.05 |

Table 5: Mean Absolute Differences (MAD) for 3-layer energy and force output blocks when the inputs are rotated, Equations (15,16). The MADs are small ($< 1\%$) when compared to the energy and force MAEs, and significantly smaller than other blocks.

OC20 IS2RE Direct Test

| Model | Energy MAE [meV] ↓ | | | | | EwT [%] ↑ | | | |
|---|---|---|---|---|---|---|---|---|---|
| | ID | OOD Ads | OOD Cat | OOD Both | Average | ID | OOD Ads | OOD Cat | OOD Both |
| Median baseline | 1750 | 1879 | 1709 | 1664 | 0.71 | 0.72 | 0.89 | 0.74 | |
| CGCNN[45] | 615 | 916 | 622 | 851 | 751 | 3.40 | 1.93 | 3.10 | 2.00 |
| SchNet[38] | 639 | 734 | 662 | 704 | 685 | 2.96 | 2.33 | 2.94 | 2.21 |
| PaiNN [36] | 575 | 783 | 604 | 743 | 676 | 3.46 | 1.97 | 3.46 | 2.28 |
| TFN ($SE_{lin}$) [5] | 584 | 766 | 636 | 700 | 672 | 4.32 | 2.51 | 4.55 | 2.66 |
| GemNet-dT[14] | 527 | 758 | 549 | 702 | 634 | 4.59 | 2.09 | 4.47 | 2.28 |
| DimeNet++[12] | 562 | 725 | 576 | 661 | 631 | 4.25 | 2.07 | 4.10 | 2.41 |
| GemNet-OC[15] | 560 | 711 | 576 | 671 | 630 | 4.15 | 2.29 | 3.85 | 2.28 |
| SphereNet[28] | 563 | 703 | 571 | 638 | 619 | 4.47 | 2.29 | 4.09 | 2.41 |
| SEGNN[5] | 533 | 692 | 537 | 679 | 610 | **5.37** | 2.46 | **4.91** | 2.63 |
| SCN | **516** | **643** | **530** | **604** | **573** | 4.92 | **2.71** | 4.42 | **2.76** |

Table 6: Results on IS2RE OC20 Test for approaches that directly predict the relaxed energies without performing relaxations and do not use auxiliary losses during training. See Tables 1 and 2 for results using relaxation based approaches to IS2RE. Results are shown for two metrics; energy Mean Absolute Error (MAE) and Energy within Threshold (EwT). The SCN model uses $L = 6$ with 16 layers and is trained for 21 epochs.

# B   OC20 IS2RE Direct Results

The task of Initial Structure to Relaxed Energy (IS2RE) may be accomplished using two approaches: 1) Use an S2EF model to relax the positions of the atoms (find local energy minima) and output the energy at the relaxed structure, or 2) directly predict the relaxed energy from the initial structure without performing the relaxation. Empirically, the first approach reported previously in the paper achieves higher accuracy (see Tables 1 and 2 in the main paper), while the second approach is more efficient both during training and inference. Although solving the same problem, the second approach of direct prediction is a fundamentally different problem than the first. The direct approach cannot take advantage of detailed position and angular information, since the initial structure only contains a rough placement of the atoms. Instead it must rely more on global knowledge of how the atoms are arranged. Thus, an approach that works well for relaxation based approaches to IS2RE might not work well for direct approaches.

As shown in Table 6, the SCN model achieves state-of-the-art results for energy MAE across all test splits for models that do not use ancillary loses [46, 18]. Energy within Threshold (EwT) is comparable to SEGNN [5]. Improvements for SCN are most pronounced for the out-of-domain splits, demonstrating its increased ability to generalize to unseen data.

# C   Implementation details

In this section, we describe additional implementation details not in the main paper.

## C.1   Varying spherical channel resolution

For atoms that are far away from each other, the messages may be computed using lower resolution spherical channels to reduce memory usage while maintaining similar accuracy. For all results in this paper, the 12 closest neighbors to each atom use the settings described in the paper. For neighbors ranked 13 to 40 by distance, $L$ is reduced by 2 with a minimum value of 4, and H is reduced by a factor of 4. Messages are not calculated for neighboring atoms with ranked distances greater than 40 or a cutoff distance greater than 8 Å.

## C.2 Point sampling on the sphere for output blocks

For both the energy and force output blocks the integrals in Equations 4 and 5 in the main paper are approximated using points samples on the sphere. Finding a set of evenly distributed points on a sphere for an arbitrary number of points remains an open problem. We use points that are approximately evenly distributed using spherical Fibonacci point sets [21]. Since the density of points does vary slightly across the sphere, we weight each point using a Gaussian weighted counting of nearby points on the unit sphere with $\sigma = 0.5$.

## C.3 PaiNN baseline

The PaiNN baseline is our reimplementation of Schütt *et al.* [36] with the difference that forces are predicted directly from vectorial features via a gated equivariant block instead of gradients of the energy output. This breaks energy conservation but is essential for good performance on OC20.

# D   Note on spherical harmonics properties

In the paper it is stated that $m = 0$ spherical harmonics are invariant to rotations about the z-axis. This is easily seen by looking at the equations for the real spherical harmonics. We show the $Y_{lm}$ equations for up to $l = 2$ below, parameterized by $\theta$ (polar) and $\phi$ (longitudinal rotation about the z-axis). Notice that all $m = 0$ spherical harmonics are only a function of $\theta$, and thus invariant to changes in $\phi$.

In Equations 4 and 5 we state that the $m = -1$ and $m = 1$ values are sine and cosine functions of $\phi$. If we integrate or fix $\theta$, we see that the values for $m = \{-1, 1\}$ do indeed take the form of Equations 4 and 5.

$$Y_{2,-2}(\theta, \phi) = \sqrt{\frac{15}{16\pi}} \sin(2\phi) \sin^2 \theta$$

$$Y_{1,-1}(\theta, \phi) = \sqrt{\frac{3}{4\pi}} \sin \phi \sin \theta \qquad Y_{2,-1}(\theta, \phi) = \sqrt{\frac{15}{4\pi}} \sin \phi \sin \theta \cos \theta$$

$$Y_{0,0}(\theta, \phi) = \sqrt{\frac{1}{4\pi}} \qquad Y_{1,0}(\theta, \phi) = \sqrt{\frac{3}{4\pi}} \cos \theta \qquad Y_{2,0}(\theta, \phi) = \sqrt{\frac{5}{16\pi}} (3\cos^2 \theta - 1)$$

$$Y_{1,1}(\theta, \phi) = \sqrt{\frac{3}{4\pi}} \cos \phi \sin \theta \qquad Y_{2,1}(\theta, \phi) = \sqrt{\frac{15}{4\pi}} \cos \phi \sin \theta \cos \theta$$

$$Y_{2,2}(\theta, \phi) = \sqrt{\frac{15}{16\pi}} \cos(2\phi) \sin^2 \theta$$

# E   Overfitting on the training dataset

Figure 5 shows the train and validation errors during training. Interestingly, we see very little overfitting on forces, but energy has significant overfitting. This may be due to the energies being a per-structure property, while the forces have more examples since they are a per-atom property. However, overfitting on the energies is still surprising given the large number of examples in the OC20 All + MD training dataset. This indicates that the SCN model can fit complex functions but could improve on its ability to better generalize through model improvements or data augmentation. Similar trends are found when trained on the OC20 2M dataset.

# F   Impact of model size

Model size has an impact on the accuracy of the SCN model. In Figure 6 we compare the accuracy and model size of SCN and GemNet-OC. For similar model sizes, we see SCN achieves better accuracy than GemNet-OC. This demonstrates that the improvement we see from SCN is more than

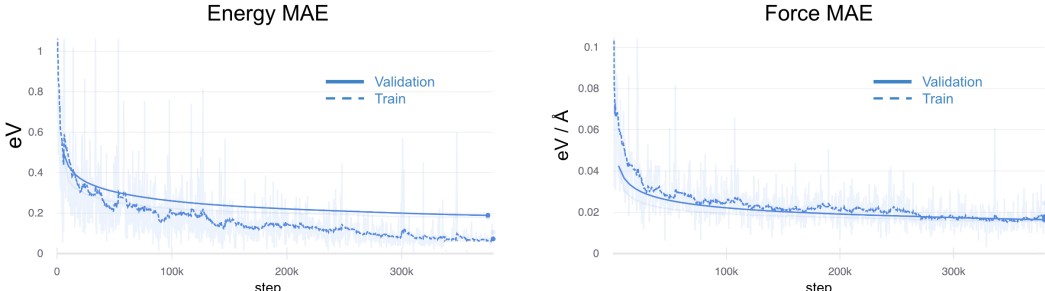

Figure 5: Plot of training and validation errors during training on the OC20All + MD dataset. The validation error is calculated on a 30k subset of the validation ID dataset. Note significant overfitting is seen for energies but not forces. Errors are smoothed using a 0.9 exponential moving average. Plot is generated from training run in Table 2

just the use of larger model sizes. In Figure 7, we see the accuracy of the SCN model improves as larger models are used on the 2M dataset. However, the accuracy for even the smallest model still outperforms other approaches reported in Table 1 in the main paper.

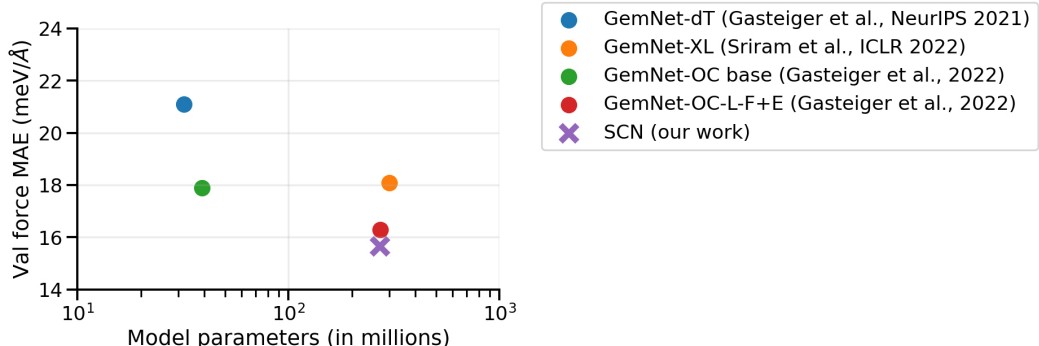

Figure 6: Plot of validation force MAE as a function of model parameters for a variety of models trained on the OC20 S2EF All dataset, including the large variants GemNet-XL and GemNet-OC-L-F+E. Note that for similar model sizes, SCN outperforms the previous state-of-the-art GemNet-OC models.

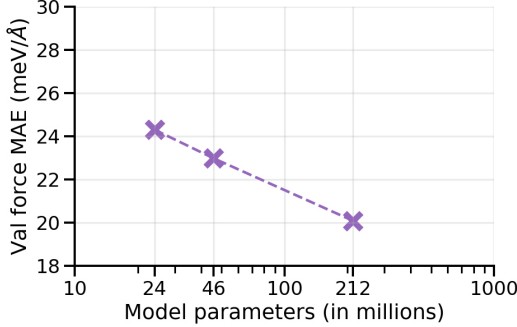

Figure 7: Plot of validation force MAE across three SCN model sizes, all models are trained on the OC20 S2EF 2M dataset.