# OpenReview forum: "Spherical Channels for Modeling Atomic Interactions"
_NeurIPS.cc/2022/Conference — NeurIPS 2022 Accept_

### Official Review · Reviewer_RB4y · 2022-07-11

**Rating:** 5
**Confidence:** 3
**Soundness:** 2 fair
**Presentation:** 3 good
**Contribution:** 2 fair

**Summary:**

The authors propose a graph neural network (GNN) they name the "Spherical Channel Network" (SCN) to map molecular configurations to atomic energies and forces. They model the atom embeddings as a set of spherical channels using spherical harmonics. The authors relax the equivariance constraint in both message passing and aggregation and demonstrate SOTA results on the Open Catalyst dataset for the prediction of energies and forces. The proposed methodology aims to replace the computationally expensive Density Functional Theory (DFT) calculations necessary to compute an estimate of atomic energies and forces.

**Questions:**

* Can the authors compute the significance of the reported results as compared to the other baseline models reported in the manuscript?
* For the non-equivariant scenario, how is the variance in the predictions affected by the number of translations in the training set?
* Have the authors tried evaluating the models on larger molecules than seen in the training set? Is there any reasoning behind outliers or failures seen during testing?

**Limitations:**

Yes limitations adequately specified.

**Strengths And Weaknesses:**

Strengths:
* The use of spherical harmonics to reason about directional information for all neighboring atoms as opposed to atom pairs.
* Apparent superior out-of-domain generalization compared to baseline models.
* Model allows for the complex mixing of different degrees of spherical harmonics enabling improvements in representation learning that can improve generalization performance.

Weaknesses:
* Relaxation of equivariance constraints can make reproducibility difficult and introduce variance in production.
* Energy is not conserved which hinders applicability to finite chemical systems in which equilibrium dynamics is concerned.
* Requires arbitrary specification of rotation around vector $d_{st}$.

---

> ### Author Response · Authors · 2022-08-02
> **Answers to reviewer specific questions.**
>
> Thanks again for your review!
>
> **Can the authors compute the significance of the reported results as compared to the other baseline models reported in the manuscript?**
>
> Note that the models trained on the OC20 All dataset are computationally expensive to train, so training multiple models to obtain variance numbers is prohibitively expensive. The standard errors of SCN OC20 S2EF validations predictions are quite small with an energy MAE standard error of 0.0034 and a force MAE standard error of 0.0001. As stated in the paper, we view a difference in energy MAE of 5 meV and force MAE of 0.5 meV/A as significant and greater than what we’d expect from differences in model initializations. The differences between SCN and GemNet-OC are significant with large 8%-11% reductions in errors.
>
> **For the non-equivariant scenario, how is the variance in the predictions affected by the number of translations in the training set?**
>
> SCN is translation invariant because of the way molecular graphs are constructed, so translations would not impact predictions. If you meant the error due to randomly rotating systems in the training set, there are empirical measurements on rotational equivariance in the supplementary section A.
>
> **Have the authors tried evaluating the models on larger molecules than seen in the training set? Is there any reasoning behind outliers or failures seen during testing?**
>
> There is a distribution of system sizes (20-200 atoms) present in the OC20 dataset, but we did not explicitly try evaluating SCN on substantially larger systems (O(1000) atoms). It is prohibitively expensive to run quantum chemistry calculations on systems with more than a couple hundred atoms — making it tough to have labeled data in that regime. Your second point is a great question, others have observed that models perform worse on certain chemistries within OC20 [ https://arxiv.org/abs/2206.02005 ]. If you would like to see SCN’s energy MAE or force MAE as a function of system size we can add that during the discussion period.

---

### Official Review · Reviewer_A9CN · 2022-07-11

**Rating:** 3
**Confidence:** 5
**Soundness:** 2 fair
**Presentation:** 2 fair
**Contribution:** 2 fair

**Summary:**

The authors present a novel deep learning architecture to predict scalars and non-conservative vector fields for applications in molecular modeling.

**Questions:**

- How is the orientation $\hat{r}$ chosen for the node embeddings? Is this choice of $\hat{r}$ independent for all atoms? How do you compute (4) efficiently?
- Do you integrate over all rotations in (4) and (5). I believe a $d\hat{r}$ is missing.
- Have you considered reporting standard results on datasets like (rev)MD17 or 3BPA?
- What is the effect of the model's size on its performance?
- Did you study the effect of the fact that the predicted forces are not conservative? Did you try building a conservative force field from your model for the energy?

**Limitations:**

The authors have adequately addressed the potential negative societal impact of their work.

**Strengths And Weaknesses:**

# Strengths

- The model architecture is simple.
- The (enormous) model seems to perform better than other (smaller) models on the OC20 task
- The model seems to be faster than other models (despite its size).

# Weaknesses

- The manuscript fails to relate the presented model to existing ones on an architectural level. While the related work section mentions other models, it is unclear what it is about SCN that other approaches should adopt to improve predictive performance and reduce computational time.
- It is unclear why the presented model outperforms existing ones on the OC20 dataset. The presented model contains up to 10x more parameters than many other (competitive) models. There is no ablation study investigating the effect of model's size.
- The model is not run on standard benchmarks such as (rev)MD17 or 3BPA. As a result, it is difficult to compare this model to existing ones and pinpoint the source of its performance.
- The model is not implementing a conservative force field. As a result, the applicability of the model to problem in computational chemistry mentioned in the abstract is severely limited. For instance, it is unsuitable as a potential in MD simulations.

---

> ### Author Response · Authors · 2022-08-02
> **Answers to reviewer specific questions.**
>
> Thanks again for your review! We responded to some of the stated weaknesses in the common response and we have tried to answer all your questions below.
>
> **How is the orientation r chosen for the node embeddings? Is this choice of $r$ independent for all atoms?**
>
> The node embeddings are functions over the entire sphere which may be sampled at different orientations $r$. When computing messages, we orient these embeddings based on edge directions with the single remaining orientation randomly sampled (line 117 original, line 119 revised). The node embeddings are initialized from an embedding based on the atom’s atomic number for $l = 0$ coefficients and the $l  \neq 0$ coefficients are set to zero. Before training, the embeddings are randomly initialized.
>
> **How do you compute (4) efficiently? Do you integrate over all rotations in (4) and (5). I believe a $dr$ is missing.**
>
> In practice, a discrete approximation of the integral is performed for equations (4) and (5) using a set of 128 evenly distributed points on the sphere (see Appendix C.2). Increasing the number of discrete samples further did not improve the final accuracies. Yes, a $dr$ is missing, thank you!
>
> **Have you considered reporting standard results on datasets like (rev)MD17 or 3BPA?**
>
> We designed SCN to be a large generalizable machine learning potential that has good performance across a large swath of chemical space. With that goal in mind, we chose OC20 as the benchmark because of its size (O(100M)) and diversity (55 elements). Optimal model architectures may differ substantially for different data regimes. For example, large transformers do not perform well in the small data regime without pre-training.  Similarly, we do not necessarily expect models that perform well on OC20 (O(100M)) to perform well on MD17 (O(1k)) and vice versa, which has been acknowledged by others [GemNet-OC]. Lastly, we note that the OC20 IS2RE task has become a popular benchmark in the community with numerous approaches on the leaderboard.
>
> **What is the effect of the model's size on its performance?**
>
> We have included two plots in the supplementary to try and help answer this and related questions. Supplementary Fig. 2 relates the OC20 performance of SCN to other large GNNs and demonstrates the success of SCN is not only due to scale. The second plot (Supplementary Fig. 3) quantifies the impact of model size on force MAE for SCN.
>
> **Did you study the effect of the fact that the predicted forces are not conservative? Did you try building a conservative force field from your model for the energy?**
>
> We did not directly study the impact of non-energy conserving force prediction. What we can say is that SCN performs state-of-the-art on the OC20 IS2RE task using an iterative relaxation approach mentioned above, which demonstrates the SCN model can iteratively find a stable local energy minima. To answer the second question, we have tried the gradient-based approach for SCN and in general it performs 10-15% worse on force predictions and 10% worse on energy predictions. For example on a 30k subset of validation ID, the same model ($l=4$, 12 layer) produces an energy MAE of 254 meV and 277 meV for direct and gradient-based (energy conserving) variants respectively. For forces, the MAEs are 22.7 and 25.7 meV/A respectively. This demonstrates that directly predicting forces is more accurate, but an energy conserving model is still competitive. Note that training an energy conserving model is also considerably slower by 4x-8x since additional gradients need to be computed.

---

> > ### Comment · Reviewer_A9CN · 2022-08-09
> > **Response to Authors**
> >
> > Thank you for your detailed response. Showing results on standard benchmarks such as (r)MD17 or 3BPA is common practice in the field - even if a novel (and potentially interesting) model does not perform well on these benchmarks. Unfortunately, the current manuscript is missing these essential results. In addition, the authors did not study the impact of the lack of energy conservation on force predictions. For these reasons, I will keep my score.

---

> > > ### Author Response · Authors · 2022-08-09
> > > **Response to Reviewer**
> > >
> > > Thank you for your response. After the rebuttal period, we allowed the energy conserving model with $l=4$ and 12 layers to fully converge. While this model is significantly slower to train (~8x), it does achieve very similar results to the model that directly estimates forces. The energy-conserving model gets 253 meV energy MAE (vs. 252 meV for direct) and 23.6 meV/A (vs. 22.5 meV/A for direct) on a 30k subset val ID. This demonstrates that even the energy-conserving variant of SCN that computes forces using the gradient of the energy with respect to the atom positions is equally effective, albeit 8x more compute-expensive. The direct model we’re comparing to here – SCN 4-12 – does produce results similar on OC20 2M to the previous state-of-the-art GemNet OC model (see Table 1 in the paper). We will add these results to the paper.
> > >
> > > While we did not optimize SCN for MD-17, and believe simpler models with more constraints are better suited for that dataset, we did train a model on it. To make an analogy, we wouldn't expect a model designed for ImageNet to perform well on MNIST and vice-versa. The results of SCN on MD-17 are below. We are happy to include these results in the camera-ready supplementary if desired.
> > >
> > > Energy (meV): Aspirin 30, Ethanol 10, Malonaldehyde 16, Naphthalene 20, Salicylic acid 19, Toluene 11, Uracil 15
> > > Forces (meV/A): Aspirin 18, Ethanol 12, Malonaldehyde 18, Naphthalene 8, Salicylic acid 17, Toluene 8, Uracil 14

---

### Official Review · Reviewer_xHpZ · 2022-07-12

**Rating:** 7
**Confidence:** 4
**Soundness:** 3 good
**Presentation:** 3 good
**Contribution:** 3 good

**Summary:**

The authors develop new kinds of deep learning models for predicting energies and forces from 3D molecular structures. Their core approach is based on representing node embeddings as spherical signals ("spherical channels" via coefficients in a spherical harmonic basis) and iteratively updating them in a message-passing process. While this is similar to some prior equivariant networks for 3D molecular structure, they make a few changes: (1) they orient the spherical signals during message passing in a way where the ij direction vector is consistent (non equivariant because the part of the rotation is underdetermined and randomized) and (2) they use more flexible pointwise processing to process the  signals by converting them back and forth to grid representations (approximately equivariant based on sampling resolution). They find that this approach gives strong empirical performance on predicting forces, energies, and relaxed energies on the Open Catalyst dataset.

**Questions:**

My main request to authors would be that they address the Loss of equivariance issue described in the Weaknesses above. It would be helpful to have further discussion on how the force and energy errors under rotation might be further mitigated and also how much it would affect compounding error for something like molecular relaxation.

UPDATE:
8/9 - I read the authors rebuttal and appreciate their additional section about message equivariance. I stand by my review and I like the paper. I think it is sufficiently different from prior works for chemistry ML, shows compelling performance, and will challenge the the field to continue developing models that can realize these benefits but with exact equivariance.

**Limitations:**

The authors adequately discuss both model and societal limitations in the discussion section.

**Strengths And Weaknesses:**

Strengths

Clear and practical ideas. While the author's modifications break rigid equivariance, they do so in an interesting way. Predictably orienting the spherical signals during message passing intuitively makes sense and makes it possible to process the spherical harmonic coefficients in a very flexible and unconstrained way (FC layers) while still having some of the rotational degrees of freedom fixed. I found the paper well-written and easy to follow. The authors adequately position with respect to prior work.

Strong evaluation results. The results are compelling, especially for the relaxed energy tasks (IS2RE). These are interesting because in some ways they would in principle require more high-order geometric reasoning and feature processing than direct energies which can often mostly be predicted from lower-order geometric features (i.e. it would need to be able to predict how energy will change upon coordinate re-configurations).

Weaknesses

Loss of equivariance. As the authors acknowledge, their final best performing models are not equivariant  but they compare to models that are equivariant by construction. I appreciate that the authors did a supplemental analysis on the errors incurred to forces and energies upon random rotation (which were non-trivially large), and even included an analysis based on summing over multiple rotation. It would be helpful to have more discussion on alternative strategies to achieving or improving equivariance, which without summing over multiple rotations seemed to give non-trivially large errors. I am curious if there would be non-randomized, canonicalization approaches to deal with the issue, for example orienting the rotation around the d_st axis in a way that points to the molecular center of mass, or something like that.

---

> ### Author Response · Authors · 2022-08-02
> **Answers to reviewer specific questions.**
>
> Thanks again for your review! We have attempted to address and expand on the loss of equivariance in a couple ways. In the revised version of our paper we included a section (2.2.1 Message Equivariance) on improving the empirically measured rotational equivarance of our message function. Additionally, in the common response we further discussed strict equivariance and provide a rationale for potentially loosening that constraint. As for your question on compounding error in molecular relaxations, we want to point out that currently SCN is state-of-the art on the IS2RE task (Table 2) via a relaxation approach. The relaxation approach entails using an S2EF trained SCN model to predict the energy and per atom forces iteratively, updating atom positions based on the forces until a convergence criteria is met, similar to what is done in a density functional theory (DFT) relaxation. So, compounding error in SCN predictions due to loss of equivariance does not have a substantial impact on relaxations compared to other models such as GemNet-OC that make invariant energy and equivariant force predictions.

---

### Author Response · Authors · 2022-08-02
**Response to common questions from reviewers.**

Thanks to all the reviewers for their constructive and thoughtful feedback. There were some common remarks between reviewers so we would like to address those here.

One of the main questions our paper poses to the community is whether or not strict equivariance is necessary? It has been assumed that strict enforcement of equivariance is imperative because it is a physically meaningful property, but what if on a practical level the constraints current equivariant architectures impose do not allow the network to train effectively on large diverse dataset like OC20? The SCN model demonstrates that strict enforcement of equivariance may not be optimal. We are not aware of any fully equivariant model that has comparable performance to SCN on OC20. One large architectural difference between SCN and other common equivariant GNNs is the way in which spherical harmonic representations interact. In many equivariant models, representations interact via tensor products followed by a gated non-linearity [TFN, NequIP, SEGNN], whereas SCN uses a unique message function and aggregates messages using an unconstrained non-linear pointwise function on the sphere. These operations are fundamentally different with SCN only encouraging equivariance but not strictly enforcing it. We strongly believe that a diversity of models and architectures will ultimately lead to more accurate and generalizable machine learning potentials for the important tasks in OC20.

Another common question/comment is about the property of energy conservation and how force predictions are made in SCN. It has been demonstrated that direct force models outperform gradient-based forces on OC20 for a variety of models [GemNet-OC, SpinConv, PaiNN]. Although it remains unclear why this is the case for the OC20 dataset, it is outside of the scope of this paper to try and answer that question. SCN follows this trend with its direct force models performing better.

It was stated that “the applicability of the model to problems in computational chemistry mentioned in the abstract is severely limited” due to SCN not having the property of energy conservation. We strongly disagree with this statement. The OC20 tasks do not require energy conservation and are prime examples of where quantum chemistry calculations — foundational tools in computational chemistry — can be augmented or accelerated for the purposes of molecular discovery. These important applications do not involve propagating systems in time at non-zero temperatures i.e. molecular dynamics where energy conservation is of particular importance. Note if energy conservation is desired, an energy conserving variant of the SCN model still performs competitively with only an increase of 10%-15% in force and energy MAE. Ideally, models would be energy conserving by construction but those that are not still have significant applicability to important real-world problems in climate, chemistry and material science.

---

### Meta-Review · Area_Chair_D4vs · 2022-08-29

**Recommendation:** Accept
**Confidence:** Less certain

**Metareview:**

The paper presents a new model with state-of-the-art results on the IS2RE task on the OpenCatalyst dataset. 2 of 3 reviewers recommended acceptance, while the 3rd reviewer raised issues with lack of baseline comparisons on small standard molecular potential energy surface datasets. I am inclined to agree with the authors on this one - if the empirical results on a large, challenging benchmark are good enough, baseline comparisons on small benchmarks are not always necessary (did the AlexNet paper include benchmarks on MNIST?) It is slightly unnerving that the results are so good with a non-equivariant model, but it sounds like the authors discuss this as well. Overall this seems like a solid, if not revolutionary, improvement in machine-learned potentials for OpenCatalyst. I recommend acceptance.

**Award:**

No

---

### Decision · Program_Chairs · 2022-09-14

Accept